# Korean Red Ginseng-Induced SIRT3 Promotes the Tom22–HIF-1α Circuit in Normoxic Astrocytes

**DOI:** 10.3390/cells12111512

**Published:** 2023-05-30

**Authors:** Hyungsu Kim, Sunhong Moon, Dohyung Lee, Jinhong Park, Chang-Hee Kim, Young-Myeong Kim, Yoon Kyung Choi

**Affiliations:** 1Department of Bioscience and Biotechnology, Konkuk University, Seoul 05029, Republic of Korea; k05042001@konkuk.ac.kr (H.K.); sunhong95047@konkuk.ac.kr (S.M.); victorydhlee@konkuk.ac.kr (D.L.); wls2137@konkuk.ac.kr (J.P.); 2Department of Otorhinolaryngology-Head and Neck Surgery, Konkuk University Medical Center, Konkuk University School of Medicine, Seoul 05030, Republic of Korea; changhee.kim@kuh.ac.kr; 3Department of Molecular and Cellular Biochemistry, School of Medicine, Kangwon National University, Chuncheon 24341, Republic of Korea; ymkim@kangwon.ac.kr

**Keywords:** SIRT3, Korean red ginseng extract, Tom22, HIF-1α, astrocytes, VEGF

## Abstract

Astrocytes play a key role in brain functioning by providing energy to neurons. Increased astrocytic mitochondrial functions by Korean red ginseng extract (KRGE) have been investigated in previous studies. KRGE administration induces hypoxia-inducible factor-1α (HIF-1α) and vascular endothelial growth factor (VEGF) in astrocytes in the adult mouse brain cortex. VEGF expression can be controlled by transcription factors, such as the HIF-1α and estrogen-related receptor α (ERRα). However, the expression of ERRα is unchanged by KRGE in astrocytes of the mouse brain cortex. Instead, sirtuin 3 (SIRT3) expression is induced by KRGE in astrocytes. SIRT3 is a nicotinamide adenine dinucleotide (NAD^+^)-dependent deacetylase that resides in the mitochondria and maintains mitochondrial homeostasis. Mitochondrial maintenance requires oxygen, and active mitochondria enhance oxygen consumption, resulting in hypoxia. The effects of SIRT3 on HIF-1α-mediated mitochondria functions induced by KRGE are not well established. We aimed to investigate the relationship between SIRT3 and HIF-1α in KRGE-treated normoxic astrocyte cells. Without changing the expression of the ERRα, small interfering ribonucleic acid targeted for SIRT3 in astrocytes substantially lowers the amount of KRGE-induced HIF-1α proteins. Reduced proline hydroxylase 2 (PHD2) expression restores HIF-1α protein levels in SIRT3-depleted astrocytes in normoxic cells treated with KRGE. The translocation of outer mitochondrial membranes 22 (Tom22) and Tom20 is controlled by the SIRT3-HIF-1α axis, which is activated by KRGE. KRGE-induced Tom22 increased oxygen consumption and mitochondrial membrane potential, as well as HIF-1α stability through PHD2. Taken together, in normoxic astrocytes, KRGE-induced SIRT3 activated the Tom22–HIF-1α circuit by increasing oxygen consumption in an ERRα-independent manner.

## 1. Introduction

Astrocytic mitochondria functions appear to be intimately linked to the regulation of aging [1,2], and secretion of angiogenic/neurogenic factors [3,4,5,6,7] under physiological conditions. Since astrocytes boost communication between the vascular and neuronal systems [6,8,9], enhanced astrocytes may contribute to dynamic neurovascular communications. The Korean red ginseng extract (KRGE) enhances astrocyte proliferation in the subventricular zone, a neurogenic area of adult mice brains [4]. KRG injection triggers neurogenesis and the differentiation of neuroblasts in the hippocampus in mice [10]. KRGE can facilitate angiogenesis in both in vivo and in vitro models by activating the glucocorticoid receptor [11].

Sirtuins (SIRT 1–7) belong to a family of importantly conserved nicotinamide adenine dinucleotide (NAD^+^)-dependent protein deacetylases. The anti-aging potential of SIRTs has also been studied [12,13]. Mitochondrial SIRT3 plays a critical role in the regulation of host mitochondrial functions under physiological conditions [14], and the expression thereof has also been detected in neurons and astrocytes obtained from mature rat brains [15]. 

Increased oxygen consumption from improved mitochondria can stabilize hypoxia-inducible factor-1α (HIF-1α) via proline hydroxylases (PHD1-3) [16,17,18]. HIF-1α and estrogen-related receptor α (ERRα) act as transcription factors for vascular endothelial growth factor (VEGF) [19,20], an effective angiogenic and neurogenic factor [21,22]. 

Healthy mitochondria have proper mitochondrial membrane potential. Nuclear-encoded proteins are first brought into the mitochondria by the translocase of the outer mitochondrial membrane (TOM) complex in the outer mitochondrial membrane [23]. Tom22 binds to the outer membrane of the mitochondria with other TOM complexes, such as Tom40 [24]. Tom20 and Tom22 are essential for maintaining the mitochondrial membrane potential in stressed astrocytes in the presence of KRGE [25].

Our previous study showed that KRGE induced HIF-1α in astrocytes. However, the effects of SIRT3 on HIF-1α-mediated mitochondrial functions induced by KRGE have not been well established. We aimed to examine the relationship between SIRT3 and HIF-1α in KRGE-treated normoxic astrocyte cells, to determine whether SIRT3-mediated mitochondrial functions can stabilize HIF-1α and lead to various biological responses, such as neurovascular regeneration (i.e., neurogenesis, angiogenesis) possibly through VEGF.

## 2. Materials and Methods

### 2.1. Materials

From SigmaAldrich (St. Louis, MO, USA), we obtained the following reagents: ethylene-diamine-tetraacetic acid (EDTA), mannitol, sucrose, MOPS, Triton X-100, phenylmethylsulfonyl fluoride (PMSF), and glycerol. From Duchefa Biochemie (Haarlem, The Netherlands), we purchased N-(2-hydroxyethyl) piperazine-N′-(2-ethanesulfonic acid) (HEPES), sodium dodecyl sulfate (SDS), NaCl, and DL-dithiothreitol (DTT). Fetal bovine serum (FBS) was purchased from Corning (Castle Rock, CO, USA). Bovine serum albumin (BSA) was purchased from US Biological Life Sciences (Salem, MA, USA). The proteinase inhibitor cocktail was purchased from Thermo Fisher Scientific (Waltham, MA, USA), and the phosphatase inhibitor cocktail from Cell Signaling Technology (Danvers, MA, USA). Phosphate-buffered saline (PBS) was obtained from Elpis Biotech (Daejeon, Republic of Korea). We used skim milk (BD Difco, BD, Franklin Lakes, NJ, USA). Korea Ginseng Corporation provided the KRGE powder. The stock solutions were made using filtered distilled water. The aliquoted stock solutions (250 mg/mL) were kept at −25 °C in the dark.

### 2.2. Animals

All mouse experiments were approved (approval number KW-181109-1) by the Animal Ethics Committee of Kangwon National University. Additionally, this study conformed to the Guide for the Care and Use of Laboratory Animals available by the United States National Institutes of Health. Seven weeks old Male C57BL/6 mice were acquired from Joong Ah Bio Inc. (Suwon, Republic of Korea) and kept in standard conditions with water and food available ad libitum for one week before experiments. 

### 2.3. Preparation of Brain Tissues for Immunohistochemistry

KRGE (0.125 or 0.25 mg/mL for 3 days) was administered using the drinking water in eight weeks old mice. The control group mice were managed by only water. For immunohistological investigations via euthanasia, mice were anesthetized using N_2_O gas and 1.5% isoflurane. Mice were transcardially perfused with 0.9% NaCl. Brains were put into optimal cutting temperature compound and then incubated in a −80 °C deep freezer. The frozen brains were sectioned into 20-μm thicknesses using a cryostat (Thermo Scientific, HM525 NX, Runcorn, UK). The section slides were incubated with 4% paraformaldehyde for 15 min and washed in 0.1% Tween 20 in PBS (PBST) three times, followed by incubation with BSA (3%) for 1 h. The section slides were then incubated with rabbit anti-HIF-1α (1:200, Novus Biologicals, Littleton, CO, USA), rabbit anti-ERRα (1:200, Novus Biologicals), rabbit anti-VEGF (1:200, Abcam, Cambridge, UK), mouse anti-GFAP antibody (1:200, BD Bioscience, Franklin Lakes, NJ, USA), mouse anti-SIRT3 (1:200, Santa Cruz Biotechnology, Dallas, TX, USA), and rabbit anti-GFAP (1:200, Thermo Fisher Scientific) in 0.1% triton X-100 in PBS at 4 °C overnight. After washing three times, the section slides were incubated in a mixture of fluorescein isothiocyanate (FITC)-conjugated donkey anti-mouse IgG (1:300, Jackson ImmunoResearch) and/or tetramethylrhodamine (TRITC)-conjugated donkey anti-rabbit IgG (1:300, Jackson ImmunoResearch, West Grove, PA, USA) for 1 h at room temperature. The tissues were then washed with PBST (0.1% Tween-20 in PBS). The brain tissues were visualized using a mounting solution (Fluoro-Gel II containing 4′, 6-diamidino-2-phenylindole (DAPI), Electron Microscopy Sciences, Hatfield, PA, USA). The stained sections were subsequently examined using a fluorescence microscope (Eclipse Ti2-U, Nikon, Tokyo, Japan). In each image, the average intensity of ten randomly chosen cells was described.

### 2.4. Primary Human Astrocyte Cell Culture

Human brain astrocytes were purchased (ACBRI371, Applied Cell Biology Research Institute, Kirkland, WA, USA). After receiving one vial of primary human astrocytes at passage 2, we stocked cells at passages 4–5 and used passages 6–10. The astrocytes were grown in DMEM (Dulbecco’s modified Eagle’s medium; HyClone, Omaha, NE, USA) with 10% FBS and 1% penicillin/streptomycin (HyClone).

### 2.5. si-RNA Transfection

At 70~75% confluency, astrocytes were transfected with small interfering ribonucleic acid (si-RNA) against a negative control (Thermo Fisher Scientific) or SIRT1, SIRT2, SIRT3, SIRT4, SIRT5, nicotinamide phosphoribosyltransferase (Nampt), ERRα, Tom20, Tom22, PHD2 (Santa Cruz Biotechnology), and HIF-1α (Dharmacon, Lafayette, CO, USA) using RNAiMax (Thermo Fisher Scientific). We used 50 nM si-RNAs except for si-PHD2 (10 nM). After approximately 16 h of recovery, the cells were incubated with KRGE (0.5 mg/mL) or water for 24 h in DMEM without FBS.

### 2.6. Immunocytochemistry

Primary human brain astrocytes were seeded in 12-well plates (Corning). After fixing with 4% formaldehyde for 15 min, the cells were washed with 0.1% [*v*/*v*] Tween 20 in PBS [PBST] for 10 min, 0.2% PBST for 10 min, and 0.1% PBST for 10 min and incubated with BSA (3%) in PBS containing 0.1% Triton X-100 (SigmaAldrich) for 1 h at room temperature. Samples in 12-well plates were then treated with rabbit anti-HIF-1α primary antibody (1:400, Novus Biologicals, Littleton, CO, USA) or mouse anti-SIRT3 primary antibody (1:300, Santa Cruz Biotechnology) in PBS containing 0.1% Triton X-100 overnight at 4 °C. After washing with 0.1% [*v*/*v*] Tween 20 in PBST, 0.2% PBST, and 0.1% PBST, each for 10 min, the secondary antibodies were diluted in PBS containing 0.1% Triton X-100 for 1 h at 23 ± 2 °C. Secondary antibodies (Jackson ImmunoResearch) were a mixture of TRITC (1:400)- and/or FITC (1:300)-conjugated donkey immunoglobulin G (IgG). After the final wash, the cells were mounted onto each well using Fluoro-Gel II containing DAPI mounting solution. The images were acquired using a fluorescence microscope (Eclipse Ti2-U; Nikon). In each image, the average intensity of ten randomly chosen cells was described.

### 2.7. MitoTracker Staining

MitoTracker^®^ Deep Red FM, a mitochondrial membrane potential-sensitive reagent, was used to image intracellular active mitochondrial levels (Thermo Fisher Scientific). The astrocytes were plated in 12-well plates and cultured until they reached 70~75% confluence. After transfection with indicated si-RNA followed by treatment of 0.5 mg/mL KRGE for 23.5 h, cells were incubated with 2 µM MitoTracker-Red for 0.5 h. PBS was used for cell washing. The stained cells were imaged using a fluorescence microscope (Eclipse Ti2eU, Nikon), and the intensity of the red color was analyzed by ImageJ. In each image, the average intensity of five randomly chosen cells was described. 

### 2.8. Western Blot Analysis

Primary human astrocytes were cultured in a 100-mm dish (Corning, Castle Rock, CO, USA). For the detection of HIF-1α (1:2000, BD Biosciences, Franklin Lakes, NJ, USA), ERRα (1:3000, Novus Biologicals), Nampt (1:4000, AdipoGen Life Sciences, San Diego, CA, USA), and β-Actin (1:8000, SigmaAldrich), we used the whole cell lysate buffer (400 mM NaCl, 10 mM HEPES [pH 7.9], 0.1 mM EDTA, 5% glycerol, 1 mM PMSF, and 1 mM DTT). RIPA buffer (Elpis Biotech) was used to detect mitochondria-localizing and other proteins. Certain amounts of protein from the cell lysates were mixed with SDS sample buffer (bromophenol blue, Tris-Cl pH 6.8, 2% SDS [*w*/*v*], 10% glycerol [*v*/*v*], and 1% β-mercaptoethanol [*v*/*v*]). Protein samples were incubated in a heating block at 98 °C for 5 min, and then separated using SDS polyacrylamide gel electrophoresis. Proteins in gels were transferred to polyvinylidene difluoride membranes (MilliporeSigma, Burlington, MA, USA), and the membranes were blockaded using Tris-buffered saline (TBS) containing 0.1% Tween 20 (TBST) and 5% skim milk. The membranes were incubated with primary antibodies in TBST at 4 °C overnight. The primary antibodies used in this research were as follows: PHD2 (1:3000, Novus Biologicals), ERRα (1:3000, Novus Biologicals), Tom20 (1:1000, Santa Cruz Biotechnology), Tom22 (1:1000, Santa Cruz Biotechnology), SIRT1 (1:1000, Santa Cruz Biotechnology), SIRT3 (1:1000, Santa Cruz Biotechnology), SIRT4 (1:2000, Thermo Fisher Scientific; 1:3000, ProteinTech, Rosemont, IL, USA), AMP-activated protein kinase α (AMPKα) (1:2000, Cell Signaling Technology, Danvers, MA, USA), p-AMPKα (Thr172) (1:2000, Cell Signaling Technology), SIRT2 (1:5000, Abcam), SIRT5 (1:1000, Santa Cruz Biotechnology), mitochondria (1:5000, Abcam), heme oxygenase-1 (HO-1, 1:1500, BD Biosciences), and β-Actin (1:8000, SigmaAldrich). Then, the membranes were washed with TBST and treated with Thermo Fisher Scientific peroxidase-conjugated secondary antibodies (1:8000) in TBST. Protein levels were detected using Elpis Biotech enhanced chemiluminescence with the proper western blot exposure apparatus (Fusion Solo, Vilber, Collégien, France).

### 2.9. Oxidative Phosphorylation (OXPHOS)

10–15 μg proteins obtained from the cell lysates using the RIPA buffer were incubated with the SDS sample buffer (10% glycerol [*v*/*v*], Tris-Cl pH 6.8, 2% SDS [*w*/*v*], 1% β-mercaptoethanol [*v*/*v*], and bromophenol blue) in a heating block at 37 °C for 5 min. After heating, protein samples were applied to western blot analysis. The anti-OXPHOS (1:8000; Abcam) antibody was exploited. The OXPHOS antibody distinguishes 20 kDa NDUFB8 (complex I; NADH: ubiquinone oxidoreductase subunit B8), 30 kDa succinate dehydrogenase complex iron-sulfur subunit B (complex II), 48 kDa UQCRC2 (complex III; ubiquinol-cytochrome c reductase core protein 2), 40 kDa MTCO1 (complex IV; cytochrome c oxidase subunit I), and 55 kDa ATP5A (complex V; ATP synthase).

### 2.10. Oxygen Consumption in Live Cells

An oxygen consumption assay tool (Cayman, Ann Arbor, MI, USA) was utilized for detecting live cell oxygen consumption. Human brain astrocytes were transfected with the indicated 50 nM si-RNAs in a 60-mm dish (Falcon). After 5 h, the cells were subjected to trypsinization and seeded in a 96-well polystyrene microplate (black colored, Corning). The media were changed to DMEM without FBS and added with water or KRGE (0.5 mg/mL) for 24 h. The oxygen sensor probe was mixed with samples in each 96-well. The “No cell” well with oxygen sensor probe was set as the blank, approximately 100 μL mineral oil was laden to each well, and the 96-well plates were evaluated using a filter combination (i.e., emission wavelength [645 nm], excitation wavelength [380 nm]) at 37 °C for 75 min (Synergy H1 Microplate Reader, BioTek, Santa Clara, CA, USA). At 45 min, the difference between the absorbance in the wells with water-treated cells and the absorbance in the wells with “no cells” was decided as “1”, and the KRGE-administered groups were calculated according to the water-treated control group.

### 2.11. Intracellular NAD^+^/NADH Measurement

Human brain astrocytes were grown in a 60-mm dish (Falcon). We used an intracellular NAD^+^/NADH quantification kit (BioVision, Waltham, MA, USA). Experiments were performed based on the manufacturer’s instructions. Primary human brain astrocytes were lysed with NAD^+^/NADH extraction buffer (800 μL) followed by freeze/thaw cycles (two times repetition). To evaluate NADH, the sample lysis (400 μL) was put into a heating block at 60 °C for 30 min. The extracted sample (50 μL) with or without heating was added to a 96-well plate (Flacon), and NAD^+^ cycling mixture (50 μL) was added. NADH developer (5 μL) was then added to the mixture, which was incubated for 1–4 h. A 450 nm wavelength was used for reading absorbance by the BioTek Epoch Microplate Spectrophotometer (Santa Clara, CA, USA).

### 2.12. Mitrochondria-Cytosol Fraction

KRGE (0.5 mg/mL) or water was put into cells in 100-mm dishes (five dishes per group) for 24 h followed by PBS washing. Cell lysis was prepared using pH 7.4 sucrose-mannitol buffer (70 mM sucrose, 210 mM mannitol, 5 mM MOPS, 1 mM EDTA), and protease inhibitor cocktail (Thermo Fisher Scientific). After homogenizing with a 27-gauge and 1/2-inch needle, the homogenates were subjected to centrifugation (800× *g* for 10 min) at 4 °C to pellet (nuclear fraction). The supernatants were further centrifuged at 10,000× *g* for 20 min at 4 °C to collect pellets containing mitochondria-enriched fraction. The pellets were washed with sucrose- mannitol buffer and were subjected to centrifugation (10,000× *g* for 20 min) at 4 °C. Then, the supernatants were put into a 3-kDa cut-off centrifugal filter (Merck Millipore, Darmstadt, Germany) and centrifugated at 4000× *g* for 2 h (cytosolic fraction). The mitochondria-enriched fraction (pellet) was lysed with 30 μL of RIPA agent, kept for 20 min on ice, and then centrifuged at 15,000× *g* for 20 min at 4 °C. After quantification using the bicinchoninic acid (BCA) protein assay reagent and copper sulfate, mitochondrial protein (8–10 μg, supernatant solution) and cytosolic protein (8–10 μg) were submitted to western blot assay. Indicated antibodies were used: mitochondria (1:5000, Abcam); SIRT3 (1:1000, Santa Cruz Biotechnology); voltage-dependent anion channel 1 (VDAC1) (1:1000, Santa Cruz Biotechnology); and GAPDH (1:1000, R&D Systems, Minneapolis, MN, USA).

### 2.13. ATP Levels in Astrocyte Cells

An BioVision ATP colorimetric assay kit was used to quantify ATP levels in human astrocytes. Cells were cultured on a 60-mm dish (Corning) and were transfected with the indicated si-RNAs followed by adding water or KRGE (0.5 mg/mL) for 24 h in DMEM without FBS. After trypsinization, the cells were moved to Eppendorf tubes. The cell pellet was lysed in ATP assay buffer (140 μL) and incubated for 5 min at room temperature. The Eppendorf tubes were centrifuged at 15,000 rpm for 5–10 min at 4 °C. Next, the supernatant was mixed with the reaction mixture reagents (1:1 ratio, total 100 μL) on 96-well plates (Corning). After incubation (0.5–2 h) without light at room temperature, absorbance (570 nm) was measured using Epoch Microplate Spectrophotometer (BioTek). Lysed cells were quantified using BCA reagent plus copper sulfate. ATP levels (nmol ATP/mg protein) in the water-treated control group were determined as “1”, and the KRGE-treated groups were calculated to the control group.

### 2.14. Protein Intensity in GFAP^+^-Astrocytes

Intensity of immunoreactivity was measured by counting ten GFAP^+^-astrocytes’ shape using ImageJ with random selection. The average of ten randomized immunoreactivity was evaluated. The average value of immunoreactivity intensities from water-treated group was decided to “1”. 

### 2.15. Data Evaluation

Quantification of protein expression by western blotting, MitoTracker staining, immunohistochemistry, and immunocytochemistry was achieved using ImageJ software (http://rsb.info.nih.gov/ij/, accessed on 15 August 2022). Comparisons were valued via one-way analysis of variance and Tukey’s post-hoc test (mean ± standard deviation) in GraphPad Prism 6. Single comparisons were performed with paired *t*-test in GraphPad Prism 6.

## 3. Results

### 3.1. KRGE Induces VEGF in Astrocytes Located near the Corpus Callosum

VEGF is an important growth factor related to regeneration (i.e., angiogenesis, neurogenesis, mitochondria biogenesis) [21,22,26,27]. We evaluated whether KRGE could induce VEGF expression in KRGE-administered mouse brain. Localization of VEGF in glial fibrillary acidic protein (GFAP)-positive astrocytes was detected near the corpus callosum approximately at bregma 1 mm, whose co-localization was clearly increased by KRGE (Figure 1a,b). Mice brains that were administered 0.25 mg/mL KRGE clearly showed VEGF expression in astrocytes compared with those administered 0.125 mg/mL KRGE (Figure 1). Thus, KRGE upregulated VEGF expression in astrocytes near the corpus callosum. 

### 3.2. KRGE Induces Astrocytic HIF-1α Protein Levels near the Corpus Callosum

Regulation of VEGF can be mediated by HIF-1α and/or ERRα [19,20]. We analyzed the expression of HIF-1α and ERRα. In 0.25 mg/mL KRGE-treated mice brains, the upregulation of HIF-1α was markedly detected (Figure 2a) in astrocytes near the corpus callosum (Figure 2b). HIF-1α immunoreactivity was almost co-localized with DAPI (nucleus marker) (Figure 2a,b). However, we were unable to find any significant difference in astrocytic ERRα expression between water and 0.25 mg/mL KRGE-administered mice (Figure 2c). ERRα immunoreactivity was almost co-localized with the nucleus (Figure 2d). These data demonstrated that astrocytic VEGF expression might stem from HIF-1α when mice were administered 0.25 mg/mL KRGE.

### 3.3. KRGE-Induced SIRT3 Regulates HIF-1α Protein Levels

HIF-1α, a key transcription factor involved in cell metabolism, affected astrocytic mitochondrial biogenesis under KRGE-treated normoxic conditions [4]. We investigated which SIRTs (namely, SIRT1, SIRT2, SIRT3, SIRT4 or SIRT5) could be involved in KRGE-induced HIF-1α protein under normoxic settings because SIRTs were linked to mitochondrial biogenesis [11,26,27]. Treatment of human astrocytes with specific si-RNA targeting for SIRT (si-SIRT) showed a reduction in each target protein (Appendix A). Among them, si-SIRT3 transfection resulted in a significant reduction of KRGE-induced HIF-1α without affecting ERRα (Figure 3a). However, KRGE-induced HIF-1α protein levels did not significantly change due to si-SIRT1 or si-SIRT2 (Figure 3a). Interestingly, diminished SIRT4 and SIRT5 significantly reduced both KRGE-induced HIF-1α and ERRα (Figure 3a). The nuclear localization of HIF-1α by KRGE was reduced by si-SIRT3 (Figure 3b,c). Because the expression of HIF-1α is closely related to that of heme oxygenase-1 (HO-1) or ERRα in HO metabolite-treated astrocytes [3,17], we examined the effects of SIRT3 on the expression of HO-1 and ERRα in KRGE-treated astrocytes. Diminished SIRT3 did not downregulate HO-1 protein levels (Figure 3d). These results suggested that SIRT3 regulated HIF-1α protein levels without affecting HO-1 or ERRα.

### 3.4. KRGE-Induced Nampt Did Not Upregulate NAD^+^/NADH Ratio or HIF-1α Proteins

Nampt is an enzyme involved in NAD^+^ production and is a co-factor of the SIRT3 enzymatic reaction [28]. Nampt-mediated NAD^+^ biosynthesis may activate NAD^+^-dependent protein deacetylases, SIRTs (i.e., SIRT3 and SIRT5) [28]. Thus, we examined Nampt expression and the NAD^+^/NADH ratio in KRGE-treated astrocytes. Despite the fact that KRGE increased Nampt protein levels (Figure 4a), different KRGE doses have no effect on the NAD^+^/NADH ratio (Figure 4b). The increased levels of HIF-1α proteins were not reduced by si-Nampt in KRGE-treated astrocytes (Figure 4c). However, the KRGE-induced ERRα protein levels were significantly reduced by si-Nampt treatment (Figure 4c). These results may suggest that KRGE upregulated HIF-1α protein in a Nampt-independent manner.

### 3.5. HIF-1α-Mediated Expression of Mitochondria, Tom20, and Tom22 in KRGE-Treated Astrocytes

Since crosstalk between HIF-1α and ERRα in astrocytes cells has been reported [17], we examined the interplay between HIF-1α and ERRα in KRGE-treated in vitro astrocytes. ERRα knockdown partly reduced KRGE-mediated HIF-1α protein levels and also showed a significant reduction in KRGE-induced mitochondria and Tom22 (Figure 5a). We then checked ERRα protein levels under HIF-1α-reduced conditions. KRGE-induced ERRα was not significantly altered by si-HIF-1α (Figure 5a). Instead, the reduction of HIF-1α in the presence of KRGE resulted in a significant reduction of mitochondria, Tom20, and Tom22 protein levels (Figure 5a). Synergistic effects of mitochondria, Tom20, and Tom22 by reductions of HIF-1α and ERRα were not observed (Figure 5a). These results demonstrated that HIF-1α could play a key role in the mitochondria mass and import systems without changing the ERRα in KRGE-treated astrocytes. Diminished ERRα or/and HIF-1α did not affect KRGE-induced SIRT3 expression (Figure 5a), indicating that SIRT3 acted as an upstream factor for HIF-1α. KRGE-mediated mitochondrial mass was partially reduced by si-Tom20 but significantly reduced by the si-Tom22 (Figure 5b). Treatment of cells with both si-Tom20 and si-Tom22 significantly reduced KRGE-mediated mitochondrial mass (Figure 5b). Taken together, KRGE-induced SIRT3 regulated HIF-1α, possibly affecting proteins involved in the mitochondria import system (i.e., Tom20, Tom22) and mitochondrial mass.

### 3.6. KRGE-Induced SIRT3 Regulates Astrocytic Mitochondrial Functions

We found that 0.25 mg/mL KRGE increased SIRT3 expression mainly in astrocytes near the corpus callosum (Figure 6a,b). We evaluated the functions of in vitro astrocytic SIRT3. When SIRT3 is processed, it transforms from its 45 kilodalton (kDa) precursor form to its 28 kDa mature form. By means of mitochondrial matrix processing peptidase and mitochondrial intermediate peptidase, SIRT3 is present in the mitochondrial matrix [29]. Our results showed that the 28 kDa SIRT3 was restricted to the mitochondria, as assessed by the mitochondria-cytosol fraction assay (Figure 6c). The KRGE treatment increased the protein levels of SIRT3 and mitochondria (detection of mitochondrial mass) in the mitochondria-enriched fraction (Figure 6c). The KRGE treatment upregulated SIRT3 immunoreactivity, which was markedly reduced by si-SIRT3 in human astrocytes (Figure 6d). In KRGE-treated astrocytes, si-SIRT3 notably downregulated Tom20 and Tom22 expression in KRGE-treated astrocytes (Figure 6e). Moreover, diminished SIRT3 expression resulted in reduced mitochondrial mass (Figure 6e), OXPHOS complexes (i.e., Complex I and Complex IV) (Figure 6f), and ATP production (Figure 6g). However, SIRT3 did not affect the activation of AMPKα and vice versa in KRGE-treated astrocytes (Appendix A). These results demonstrated that KRGE-treated astrocytes played a key role in mitochondrial function through SIRT3 without affecting AMPKα activation.

### 3.7. KRGE Induces Astrocytic HIF-1α by Enhancing Tom22-Mediated Oxygen Consumption

The relationship between the Tom20/Tom22 complex and HIF-1α was investigated. HIF-1α protein was elevated by KRGE while being considerably decreased by si-Tom22 but not si-Tom20 (Figure 7a). Increased mitochondrial functions could stabilize HIF-1α because of upregulated oxygen consumption. Therefore, we observed the oxygen consumption, and our results showed that KRGE-induced oxygen consumption was blocked by si-Tom22, but not by si-Tom20 (Figure 7b). In the presence of KRGE, si-HIF-1α reduced the immunoreactivity of Tom22 while si-Tom22 reduced the immunoreactivity of HIF-1α (Figure 7c,d), indicating a possible interaction between the two proteins. The mitochondrial membrane potential was detected using the MitoTracker assay when cells were treated with si-Tom22 or si-HIF-1α. KRGE enhanced mitochondrial membrane potential, which was significantly reduced by si-Tom22 or si-HIF-1α (Figure 7e). Our findings demonstrated the crucial activities of the Tom22-HIF-1α circuit in the mitochondrial functions of KRGE-treated astrocytes (i.e., oxygen consumption and mitochondrial membrane potential).

### 3.8. KRGE-Mediated SIRT3 and Tom22 Stabilize HIF-1α Protein through PHD2

In the presence of KRGE, si-Tom22-mediated reduction of HIF-1α protein was recovered by co-knockdown of PHD2 (Figure 8a). During normoxia, PHD2 is the key oxygen sensor in the degradation of HIF-1α using oxygen [30,31]. Similar to Figure 8a, the si-SIRT3-mediated reduction of HIF-1α protein was stabilized by the co-knockdown of PHD2 (Figure 8b). These results suggested that KRGE upregulated mitochondrial SIRT3 and Tom22, leading to mitochondrial activation (i.e., oxygen consumption) and consequent inhibition of HIF-1α degradation.

## 4. Discussion

KRGE has been known to activate mitochondria functions which can be required for regeneration and anti-aging effects [1,4,11,32,33,34]; however, it is still unclear what exactly caused the active mitochondrial states it generated. Our previous data showed that an important transcription factor for cell survival and metabolism, HIF-1α, plays an essential role in KRGE-mediated astrocytic mitochondrial biogenesis [4]. Our novel findings suggested that mitochondrial astrocytic SIRT3 is an upstream factor for HIF-1α and that the SIRT3–HIF-1α axis regulates KRGE-mediated Tom22 expression. Notably, Tom22 is a crucial factor in mitochondrial oxygen consumption in KRGE-treated astrocytes, leading to the stabilization of the HIF-1α protein. Therefore, KRGE boosted the astrocytic mitochondrial function via the SIRT3-mediated Tom22–HIF-1α circuit, leading to the enhancement of the regenerative factor, VEGF (Figure 8c).

In our observation, HIF-1α may be detected in neurons as well as astrocytes; in addition, we assume that KRGE may increase the HIF-1α expression in neurons. Neuron-specific inactivation of the HIF-1α increases brain injury in a mouse model of transient focal cerebral ischemia [35]. In embryonic neural stem cells, deletion of the HIF-1α gene leads to a decrease in self-renewal of neural stem cells, paralleled by an enhancement in neuronal differentiation [36]. Therefore, KRGE may function in neuronal cells by regulating HIF-1α. In this study, the relative prevalence of SIRT3 and VEGF expression in KRGE-administered mice brains can be higher in astrocytes than in neurons localized above the corpus callosum. Thus, we focus on astrocytic VEGF expression through SIRT3–HIF-1α axis.

Nampt-mediated NAD^+^ biosynthesis is essential for SIRT3 activation [28]. KRGE-induced ERRα expression was downregulated by Nampt knockdown. However, diminished Nampt expression did not downregulate HIF-1α protein levels in KRGE-treated astrocytes. In normoxic conditions, astrocytes treated with KRGE did not result in an increase in the NAD^+^/NADH ratio. An increase in SIRT3 activation may cause a reduction in NAD^+^ levels in the mitochondria because the deacetylation activity performed by SIRT3 converts NAD^+^ into nicotinamide and acetyl ADP-ribose [37]. We hypothesize that KRGE-induced SIRT3 activation may lead to NAD^+^ reduction, even though KRGE produces NAD^+^ by Nampt. Our findings suggested that Nampt can regulate ERRα in a SIRT3-independent manner and that SIRT3 can regulate HIF-1α in a Nampt-independent manner.

How does KRGE increase the expression of SIRT3? Ginsenoside Rg1 may increase SIRT3 expression at both the mRNA and protein levels in HSCs/HPCs cells of an aging rat model [38]. In rat retinal capillary endothelial cells, Rb1 recovers the protein expression of SIRT1 and SIRT3 during high glucose-induced endothelial damage [39]. Rb1 also increases SIRT activity, which is decreased by high glucose [39]. In this study, KRGE contains 0.83 mg/g ginsenoside Rg1 and 5.52 mg/g ginsenoside Rb1. Therefore, we assume that KRGE increases the expression of SIRT3 possibly through specific ginsenoside(s)-mediated *SIRT3* gene expression. 

Energy production is important for brain homeostasis and function. Our previous study demonstrated that KRGE-induced ATP production could be downregulated by si-HIF-1α [4] or si-Tom22 [34]. In this study, KRGE-induced ATP production was completely inhibited by si-SIRT3. KRGE-induced OXPHOS complexes I and IV were reduced with lower SIRT3 expression, indicating that SIRT3 may be implicated in KRGE-mediated mitochondrial OXPHOS and ATP synthesis.

The relationship between HIF-1α and Tom22 in normoxic astrocytes has not been well established. Here, Tom22 may act as an oxygen sensor by stimulating oxygen consumption and HIF-1α stabilization. In contrast, Tom20 did not affect the protein stability of HIF-1α, even though Tom20 knockdown reduced ATP production in KRGE-treated astrocytes [34]. Since our data demonstrate that HIF-1α is mainly expressed in the nucleus of brain tissues, we assume that Tom22 regulates HIF-1α via an indirect pathway. The SIRT3-Tom22 axis regulated the mitochondrial oxygen consumption, leading to enhanced HIF-1α stabilization through PHD2 inactivation due to oxygen depletion in KRGE-treated astrocytes.

The association between Tom22 and VDAC (known as Por1 in yeast) is important to regulate mitochondrial protein gate assembly [40]. We speculate that the Tom22–VDAC complex-related import system, including Tom40, may be involved in mitochondrial functions. In recent reports, HIF-1α binds with mitochondrial protein VDAC1 [41]. Under normoxia, transcriptionally inactive forms of unmodified HIF-1α or its C-terminal domain alone can be targeted to mitochondria, stimulating production of a C-terminally truncated active form of VDAC1 [41]. Thus, we cannot exclude the possible interaction between HIF-1α and Tom22 in the mitochondrial outer membrane. 

Interestingly, the Tom22–HIF-1α circuit influenced the Tom20 expression. As mitochondrial mass can be clearly regulated by the combination of Tom20 and Tom22, the HIF-1α-mediated increase in mitochondrial mass can be modulated by both Tom22 and Tom20. The SIRT3–HIF-1α axis is a critical regulator of Tom20 and Tom22 without affecting ERRα expression under KRGE-treated normoxic conditions.

The peroxisome-proliferator-activating receptor-γ coactivator-1α (PGC-1α)–ERRα pathway is a key regulator for astrocytic mitochondrial biogenesis [3]. The PGC-1α–ERRα pathway also induces HIF-1α stability [17], possibly facilitating VEGF expression [20]. KRGE upregulated ERRα protein levels; however, SIRT3 did not alter ERRα expression. Instead, SIRT1 affected physiological brain functions, including mitochondrial biogenesis in an HIF-1α-independent manner, but in an ERRα-dependent manner [34]. Moreover, SIRT4 and SIRT5 regulated both HIF-1α and ERRα. Thus, we cannot rule out the functions of other SIRTs (i.e., SIRT4 and SIRT5) and PGC-1α in KRGE-mediated physiological regenerative effects. Hence, we can investigate the beneficial effects of various SIRTs on KRGE-mediated neurovascular metabolism in the future.

## 5. Conclusions

In conclusion, KRGE enhanced astrocytic VEGF, HIF-1α, and SIRT3 expression near the corpus callosum; however, the precise mechanism by which this occurs is not yet completely understood. According to our novel research, KRGE increased the expression of SIRT3 in astrocytic mitochondria, and SIRT3 promoted mitochondrial biogenesis and ATP synthesis, increasing oxygen consumption. These results triggered the Tom22–HIF-1α circuit through mitochondria-nucleus crosstalk, possibly leading to VEGF upregulation.

## Figures and Tables

**Figure 1 cells-12-01512-f001:**
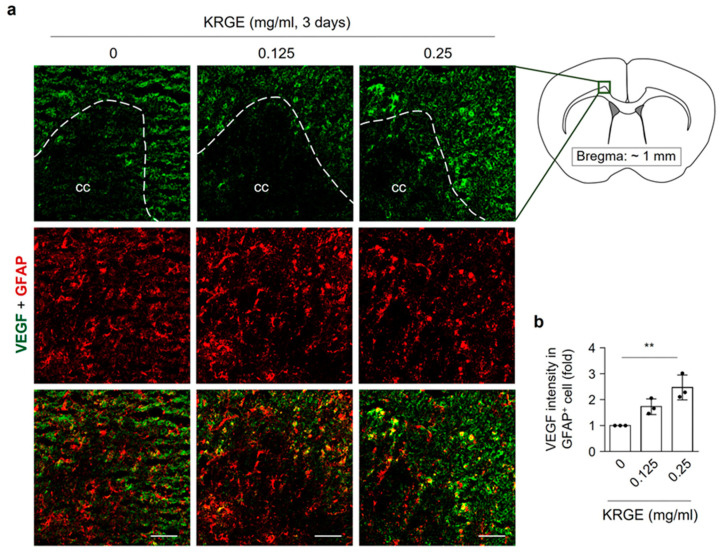
KRGE induced VEGF expression in astrocytes located above the corpus callosum. (**a**) KRGE (0.125 mg/mL or 0.25 mg/mL for 3 days) was added into drinking water for mice. Expressive images of VEGF (green) and GFAP (red) are shown located above corpus callosum (rectangle) of the brain approximately at bregma 1 mm (*n* = 3 per group). Scale bar = 50 μm. (**b**) VEGF intensity in GFAP-positive (GFAP^+^) cells of brains (**a**) was measured using Image J (*n* = 3 animals per group). Scale bar = 20 μm. ** *p* < 0.01. cc, corpus callosum.

**Figure 2 cells-12-01512-f002:**
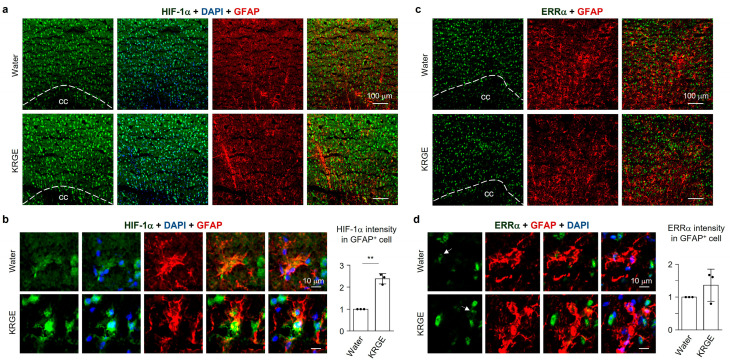
KRGE (0.25 mg/mL) induced astrocytic HIF-1α protein above the corpus callosum. (**a**) Representative images of HIF-1α (green), DAPI (blue), and GFAP (red) are shown located above the corpus callosum of the brain approximately at bregma 1 mm (*n* = 3 per group). Scale bar = 100 μm. (**b**) Magnified images of HIF-1α (green), DAPI (blue), and GFAP (red) are shown. HIF-1α intensity in GFAP^+^ cells of brains (**a**) was quantified using Image J (*n* = 3 animals per group). Scale bar = 10 μm. (**c**) Representative images of ERRα (green) and GFAP (red) are shown located above the corpus callosum of the brain approximately at bregma 1 mm (*n* = 3 per group). Scale bar = 100 μm. (**d**) Magnified images of ERRα (green), DAPI (blue), and GFAP (red) are shown. ERRα intensity in GFAP^+^ cells of brains (**c**) was quantified using Image J (*n* = 3 animals per group). Scale bar = 10 μm. ** *p* < 0.01. cc, corpus callosum.

**Figure 3 cells-12-01512-f003:**
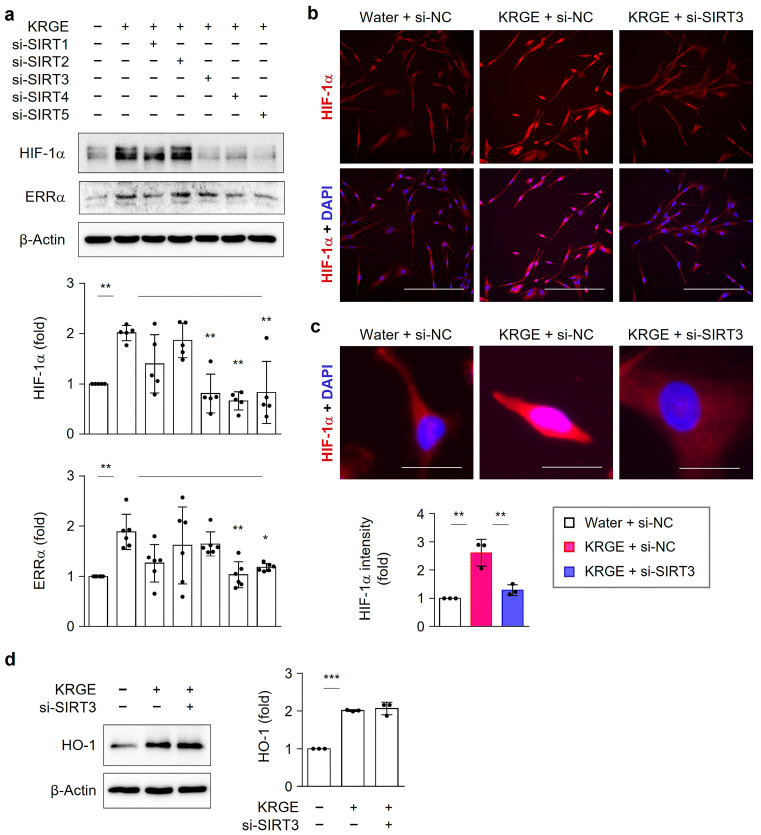
KRGE-induced SIRT3 regulates HIF-1α protein levels. (**a**–**c**) Human astrocytes were transfected with indicated si-RNA and exposed to KRGE (0.5 mg/mL) for 24 h. (**a**) Selected protein levels in cell lysate with whole cell lysate buffer were determined via western blot (*n* = 5–6). (**b**) Representative images of HIF-1α (red) and DAPI (blue) were shown (*n* = 3 per group). Scale bar = 200 μm. (**c**) Magnified images were shown. Scale bar = 20 μm. HIF-1α intensity in astrocytes was quantified using Image J (*n* = 3 independent cell cultures). (**d**) Astrocytes were transfected with indicated si-SIRT3 and subjected to KRGE (0.5 mg/mL) for 24 h. HO-1 (*n* = 3) in cell lysate with RIPA buffer were identified via western blot analysis. * *p* < 0.05; ** *p* < 0.01; *** *p* < 0.001.

**Figure 4 cells-12-01512-f004:**
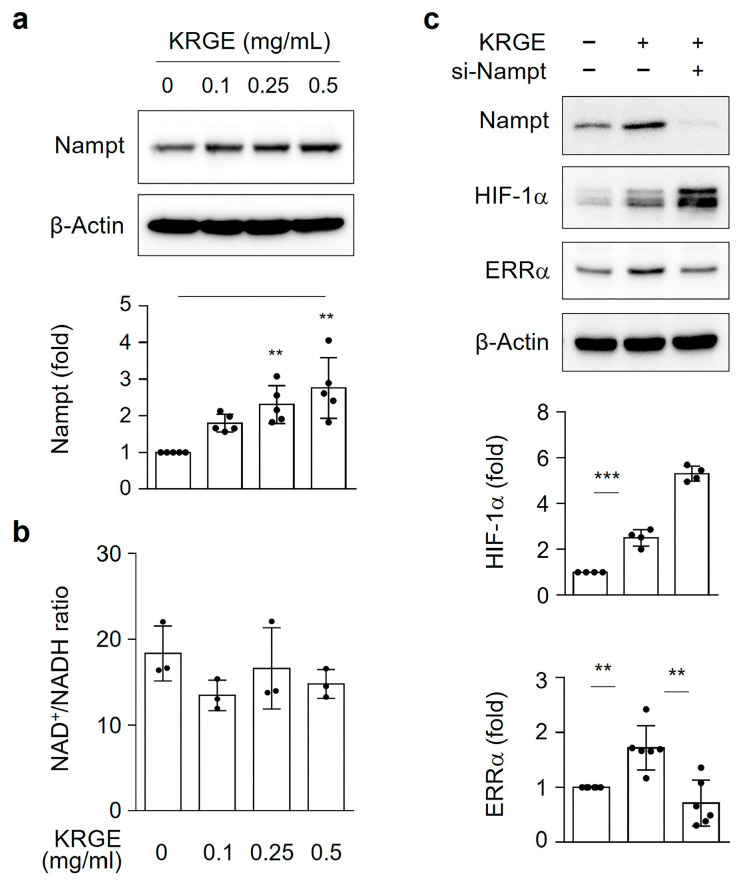
KRGE-induced Nampt did not upregulate NAD^+^/NADH ratio or HIF-1α proteins. (**a**,**b**) Various doses of KRGE were administered to human astrocytes for 24 h. (**a**) Selected protein levels were determined with quantification (*n* = 5). (**b**) NAD^+^/NADH ratio was determined (*n* = 3). (**c**) Brain astrocytes were transfected with control or indicated siRNAs and subjected to KRGE (0.5 mg/mL) for 24 h. Target protein levels (HIF-1α [*n* = 4], ERRα [*n* = 6]) were assessed by western blot. ** *p* < 0.01; *** *p* < 0.001.

**Figure 5 cells-12-01512-f005:**
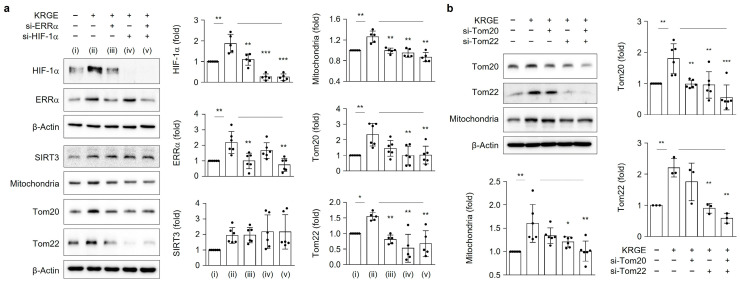
HIF-1α-mediated expression of mitochondria, Tom20, and Tom22 in KRGE-treated in vitro astrocytes. (**a**,**b**) Human brain astrocytes were transfected with indicated si-RNA and added with KRGE (0.5 mg/mL) for 24 h. * *p* < 0.05; ** *p* < 0.01; *** *p* < 0.001. (**a**) Protein levels were analyzed using western blot and quantified (*n* = 5–6). (i) water + negative control si-RNA; (ii) KRGE + negative control si-RNA; (iii) KRGE + si-ERRα; (iv) KRGE + si-HIF-1α; (v) KRGE + si-ERRα + si-HIF-1α. (**b**) Protein levels were determined using western blot and quantified (*n* = 3–6 independent cell cultures).

**Figure 6 cells-12-01512-f006:**
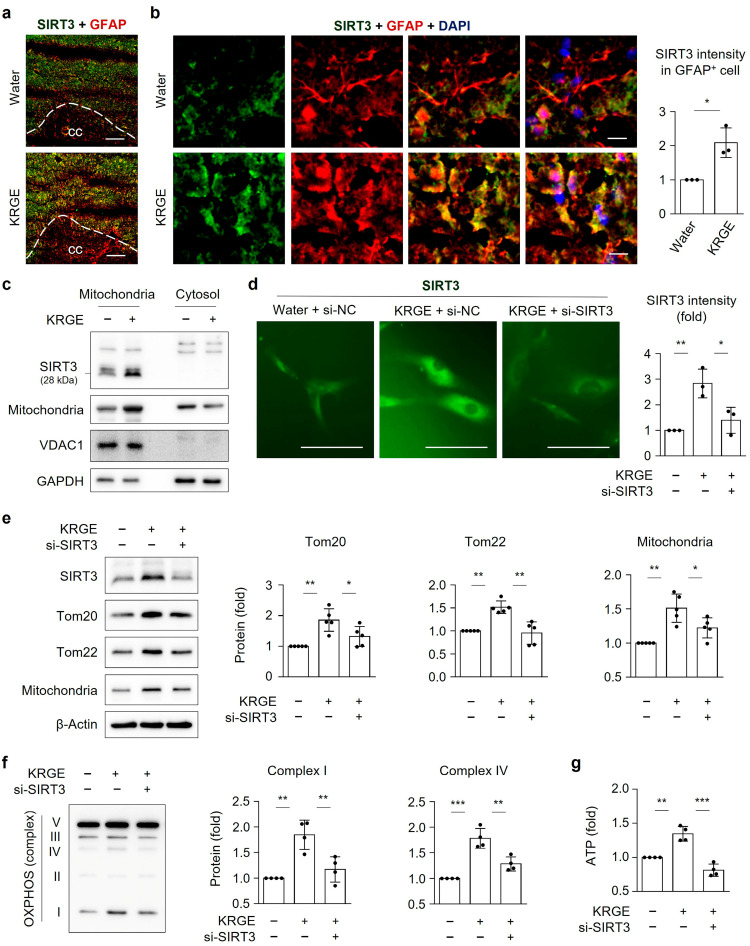
KRGE-induced SIRT3 regulates astrocytic mitochondrial functions. (**a**) Representative images of SIRT3 (green) and GFAP (red) in water- or KRGE (0.25 mg/mL)-administered mice brains (*n* = 3 per group). Scale bar = 100 μm. (**b**) Magnified images of SIRT3 (green), GFAP (red) and DAPI (blue) of brains. Scale bar = 10 μm. SIRT3 intensity in GFAP^+^ cells was quantified using Image J (*n* = 3 animals per group). cc, corpus callosum. (**c**) Proteins were obtained from cellular fractions separated into cytosol and mitochondria and detected by western blot (*n* = 3). (**d**–**g**) Astrocytes were transfected with indicated si-RNA and subjected to KRGE (0.5 mg/mL) for 24 h. (**d**) Representative images of SIRT3 (green) were shown. SIRT3 intensity in astrocytes was quantified using Image J (*n* = 3 independent cell cultures). Scale bar = 50 μm. (**e**) Protein levels in cell lysate were assessed by western blot (*n* = 5 independent cell cultures). (**f**) OXPHOS levels were detected by western blot and quantified (*n* = 4 independent cell cultures). (**g**) ATP levels were detected (*n* = 4 independent cell cultures). * *p* < 0.05; ** *p* < 0.01; *** *p* < 0.001.

**Figure 7 cells-12-01512-f007:**
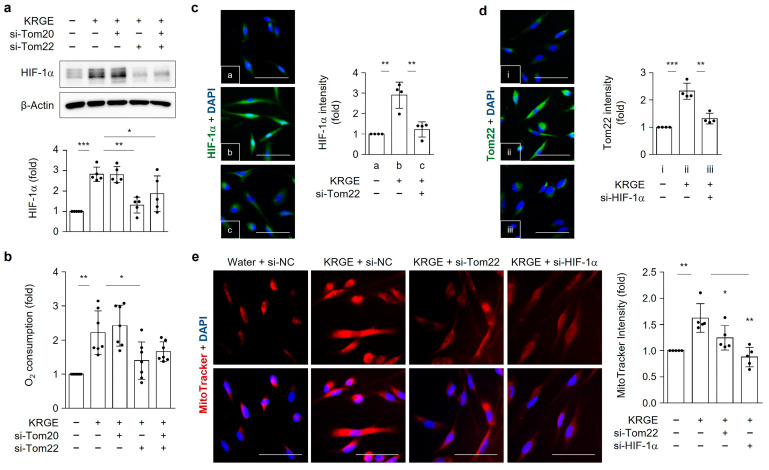
KRGE induces in vitro astrocytic HIF-1α by enhancing Tom22-mediated oxygen consumption. (**a**–**d**) Cells were transfected with indicated si-RNA and subjected to KRGE (0.5 mg/mL) for 24 h. (**a**) Selected protein levels were analyzed using western blotting and quantified (*n* = 5). (**b**) Oxygen consumption rate was determined (*n* = 7). (**c**) Representative images of HIF-1α (green) and DAPI (blue) were shown (*n* = 4 per group). HIF-1α intensity in astrocytes was quantified using Image J (*n* = 4 independent cell cultures). a, water + si-NC (negative control); b, KRGE + si-NC; c, KRGE + si-Tom22. Scale bar = 50 μm. (**d**) Descriptive images of Tom22 (green) and DAPI (blue) were shown (*n* = 4 per group). Tom22 intensity in astrocytes was quantified using Image J (*n* = 4 independent cell cultures). i, water + si-NC; ii, KRGE + si-NC; iii, KRGE + si-HIF-1α. Scale bar = 50 μm. (**e**) Images of MitoTracker (red) and DAPI (blue) were demonstrated and quantified using relative intensity (*n* = 5 per group). Scale bar = 50 μm. * *p* < 0.05; ** *p* < 0.01; *** *p* < 0.001.

**Figure 8 cells-12-01512-f008:**
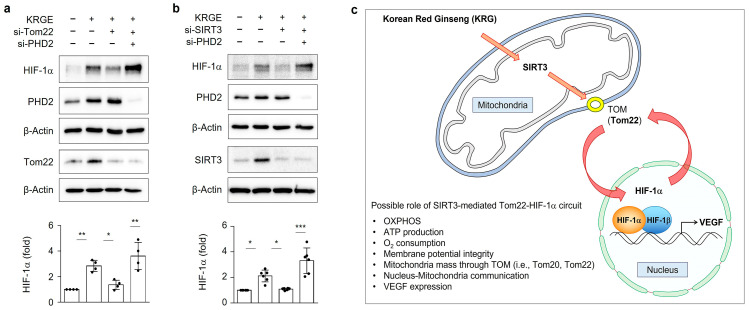
KRGE-mediated SIRT3 and Tom22 stabilize HIF-1α protein through PHD2. (**a**,**b**) Cells were transfected with indicated si-RNA and subjected to KRGE (0.5 mg/mL) for 24 h. Indicated protein levels analyzed using western blotting with quantification (*n* = 4–6 independent cell cultures). * *p* < 0.05; ** *p* < 0.01; *** *p* < 0.001. (**c**) Schematic figure showing molecular mechanisms of the present study. TOM, translocase of outer mitochondrial membrane; OXPHOS, oxidative phosphorylation; VEGF, vascular endothelial growth factor.

## Data Availability

The data presented in this study are contained within the article. Original data will be made available on request from the corresponding author.

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
