# Peer review of "Korean Red Ginseng-Induced SIRT3 Promotes the Tom22–HIF-1α Circuit in Normoxic Astrocytes"

_cells, 2023, doi:10.3390/cells12111512_

Round 1
Reviewer 1 Report
This study provides valuable insight into the potential therapeutic mechanism of a ginseng extract on mitochondrial and cell health. For the most part, the authors convincingly demonstrate that KRGE acts via mitochondrial SIRT3 to affect HIF-1alpha and its downstream targets. My enthusiasm for the study is somewhat diminished by the almost exclusively in vitro nature of the experiments (outside of some correlative immunohistochemistry from brain slices) as well as a lack of a clear rationale for the focus on astrocytes.
Major issues:
1) There is a lack of rationale for focusing this study on astrocytes. In the immunohistochemical studies (Figures 1 and 2), there are significant portions of the VEGF and HIF-1alpha signals that do not colocalize with GFAP, indicative of a response that is not astrocyte-specific. Would the findings from this study hold true if the culture manipulations were performed in neurons or even another glial cell type (i.e. microglia)?
2) Speaking of lack of rationale, the authors do not provide a strong one for the use of KRGE. In the discussion, they reference "regenerative effects on human health", but this is not explored with any detail.
3) Throughout the paper (e.g. figures 1, 2, 3B/C, 6A/B/D, 7C/D/E), the authors include experimental findings from imaging-based methods but do not provide any quantitative assessment. The authors may have omitted these analyses because they found the changes to be obvious, but that decision should be left to the audience.
4) The authors used si-RNA to knockdown various targets throughout the study. Among these, the SIRT4 (Supplementary Figure 1B) and Tom20 (Figure 5B) were not very convincing. For Tom20, this is particularly important because the authors make the claim that KRGE acts via Tom22 and not Tom20; if the Tom20 si-RNA was not working properly, this claim cannot be made effectively.
Minor issues:
1) Please provide rationale for the live animal dose of KRGE used.
2) How does the live animal KRGE dose (0.125 or 0.25 mg/mL, 3 day oral dosing) compare to the in vitro dose (0.5 mg/mL, 24 hr) in terms of direct effects on the target cells?
Author Response
Reviewer #1
This study provides valuable insight into the potential therapeutic mechanism of a ginseng extract on mitochondrial and cell health. For the most part, the authors convincingly demonstrate that KRGE acts via mitochondrial SIRT3 to affect HIF-1alpha and its downstream targets. My enthusiasm for the study is somewhat diminished by the almost exclusively in vitro nature of the experiments (outside of some correlative immunohistochemistry from brain slices) as well as a lack of a clear rationale for the focus on astrocytes.
Major issues:
#1. There is a lack of rationale for focusing this study on astrocytes. In the immunohistochemical studies (Figures 1 and 2), there are significant portions of the VEGF and HIF-1alpha signals that do not colocalize with GFAP, indicative of a response that is not astrocyte-specific. Would the findings from this study hold true if the culture manipulations were performed in neurons or even another glial cell type (i.e. microglia)?
Response #1: That is an important suggestion, and we agree your observation. HIF-1a expression within the brain is not restricted to astrocytes. HIF-1a may be detected in neurons; in addition, we think KRGE may increase the HIF-1a levels in neurons.
Neuron-specific inactivation of the HIF-1a increased brain injury in a mouse model of transient focal cerebral ischemia [1]. In embryonic neural stem cells, deletion of the HIF-1a gene led to a decrease in self-renewal of neural stem cells, paralleled by an increase in neuronal differentiation [2]. Therefore, KRGE may have function in neuronal cells by regulating HIF-1a.
- 1. Baranova, O.; Miranda, L. F.; Pichiule, P.; Dragatsis, I.; Johnson, R. S.; Chavez, J. C., Neuron-specific inactivation of the hypoxia inducible factor 1 alpha increases brain injury in a mouse model of transient focal cerebral ischemia. J Neurosci 2007, 27, (23), 6320-32.
- 2. Vecera, J.; Prochazkova, J.; Sumberova, V.; Panska, V.; Paculova, H.; Lanova, M. K.; Masek, J.; Bohaciakova, D.; Andersson, E. R.; Pachernik, J., Hypoxia/Hif1alpha prevents premature neuronal differentiation of neural stem cells through the activation of Hes1. Stem Cell Res 2020, 45, 101770.
If we could establish cell cultures such as neurons and microglia, we would like to do experiments about the relationship HIF-1a and VEGF in KRGE-treated conditions per your valuable comment.
In this study, however, the relative prevalence of SIRT3 and VEGF expression in KRGE-administered mice brains can be higher in astrocytes than in neurons localized above the corpus callosum. Although we did not show data, we checked SIRT1, SIRT2, SIRT3, SIRT4 and SIRT5 in KRGE-administered mice brains. Among them, the expression of SIRT3 was coincided with GFAP positive astrocytes in near the corpus callosum. Other SIRTs (i.e., SIRT1, SIRT2, SIRT4, SIRT5) might be not expressed in GFAP positive astrocytes localized above the corpus callosum.
Also, we checked SIRT3 expression in neuronal lineage cells (i.e., adult neural stem cells) and in glia cells (i.e., astrocytes) (bellowed figure). Direct treatment of neural stem cells with KRGE (0, 0.1, 0.25, 0.5 mg/ml) did not increase SIRT3 expression. Instead, KRGE induced SIRT3 expression in astrocytes in dose-dependent manner. Thus, we hypothesized that KRGE could increase astrocytic SIRT3.
Taken together, we investigated the role of astrocytic SIRT3 in HIF-1a-mediated mitochondrial functions and VEGF expression in this study. We found that SIRT3 could significantly regulate HIF-1a stability through Tom22-mediated mitochondrial functions such as O2 consumption, OXPHOS and ATP production.
We added them in the Discussion (Line 451-459).
Line 451-459: In our observation, HIF-1a may be detected in neurons as well as astrocytes; in addition, we assume that KRGE may increase the HIF-1a levels in neurons. Neuron-specific inactivation of the HIF-1a increases brain injury in a mouse model of transient focal cerebral ischemia [35]. In embryonic neural stem cells, deletion of the HIF-1a gene leads to a decrease in self-renewal of neural stem cells, paralleled by an increase in neuronal differentiation [36]. Therefore, KRGE may have function in neuronal cells by regulating HIF-1a. In this study, the relative prevalence of SIRT3 and VEGF expression in KRGE-administered mice brains can be higher in astrocytes than in neurons localized above the corpus callosum. Thus, we focus on astrocytic VEGF expression through SIRT3–HIF-1a axis.
#2. Speaking of lack of rationale, the authors do not provide a strong one for the use of KRGE. In the discussion, they reference "regenerative effects on human health", but this is not explored with any detail.
Response #2: We changed the Discussion per your valuable comment.
Line 439-441: KRGE has been known to activate mitochondria functions which can be required for regeneration and anti-aging effects [1, 4, 11, 30-32]; however, it is still unclear what exactly caused the active mitochondrial states it generated.
We also changed the Introduction.
Line 42-50: Astrocytic mitochondria functions appear to be intimately linked to regulation of aging [1, 2], and secretion of angiogenic/neurogenic factors [3-7] under physiological conditions. Since astrocytes boost communication between the vascular and neuronal systems [6, 8, 9], enhanced astrocytes may contribute to dynamic neurovascular communications. Korean red ginseng extract (KRGE) promotes astrocyte proliferation in the subventricular zone, a neurogenic area of adult mice brains [4]. KRG administration increases neurogenesis and the differentiation of neuroblasts in the hippocampus in mice [10]. KRGE can induce angiogenesis both in vivo and in vitro models through activating glucocorticoid receptor [11].
#3. Throughout the paper (e.g. figures 1, 2, 3B/C, 6A/B/D, 7C/D/E), the authors include experimental findings from imaging-based methods but do not provide any quantitative assessment. The authors may have omitted these analyses because they found the changes to be obvious, but that decision should be left to the audience.
Response #3: We added quantification graph in Figure 1c, 2b, 2d, 3b, 6b, 6d, 7c, and 7d.
#4. The authors used si-RNA to knockdown various targets throughout the study. Among these, the SIRT4 (Supplementary Figure 1B) and Tom20 (Figure 5B) were not very convincing. For Tom20, this is particularly important because the authors make the claim that KRGE acts via Tom22 and not Tom20; if the Tom20 si-RNA was not working properly, this claim cannot be made effectively.
Response #4: Our finding suggests that Tom22-HIF-1a circuit, which can regulate Tom20 expression. In other words, si-Tom22 downregulates Tom20 expression because Tom22-HIF-1a circuit regulates Tom20 expression. We summarized this finding bellowed. In addition, we quantified the Tom20 expression, and we found significant reduction by si-Tom20 or/and si-Tom22 transfection. We added this quantification data in Figure 5b.
In case of SIRT4, it was difficult to detect SIRT4 bands; thus, we purchased another company’s antibody. After several trials for establishing antibody condition, we got significant reduction of SIRT4 by si-SIRT4 transfection.
We added this data in Supplementary Figure 1b.
Minor issues:
#1. Please provide rationale for the live animal dose of KRGE used.
Response #1: Our previous study, we used same dose of KRGE [3] with consideration of other references [4, 5]. Adam, G. O. et al. [4] used Sprague-Dawley rats weighing (220–250 g). They received 2ml (0.66 mg/ml) of KRGE by oral gavage daily. Total amount was 1.32 mg/2 ml for one day. If we count them for 3 days, it would be 3.96 mg (1.32 mg x 3 days). Jun, Y. L. et al. [5] used C57 BL/6 mice. They received 10 mg/kg or 100 mg/kg KRGE for one day.
We administered 0.125 mg/ml or 0.25 mg/ml for 3 days. We assumed that the average daily consumption of water for an adult 25 g mouse could be 4 ml. Total amount of KRGE was 1.5 mg or 3 mg. That is corresponding with 20 mg/kg or 40 mg/kg, respectively.
- 3. Moon, S.; Kim, C. H.; Park, J.; Kim, M.; Jeon, H. S.; Kim, Y. M.; Choi, Y. K., Induction of BVR-A Expression by Korean Red Ginseng in Murine Hippocampal Astrocytes: Role of Bilirubin in Mitochondrial Function via the LKB1-SIRT1-ERRalpha Axis. Antioxidants (Basel) 2022, 11, (9).
- 4. Adam, G. O.; Kim, G. B.; Lee, S. J.; Lee, H.; Kang, H. S.; Kim, S. J., Red Ginseng Reduces Inflammatory Response via Suppression MAPK/P38 Signaling and p65 Nuclear Proteins Translocation in Rats and Raw 264.7 Macrophage. Am J Chin Med 2019, 47, (7), 1589-1609.
- 5. Jun, Y. L.; Bae, C. H.; Kim, D.; Koo, S.; Kim, S., Korean Red Ginseng protects dopaminergic neurons by suppressing the cleavage of p35 to p25 in a Parkinson's disease mouse model. J Ginseng Res 2015, 39, (2), 148-54.
#2. How does the live animal KRGE dose (0.125 or 0.25 mg/mL, 3 day oral dosing) compare to the in vitro dose (0.5 mg/mL, 24 hr) in terms of direct effects on the target cells?
Response #2: The total amount of KRGE (0.25 mg/ml) consumed by the mice for 3 days was approximately 3 mg. We assumed that the average daily consumption of water for an adult 25 g mouse could be 4 ml. The total amount of KRGE (0.5 mg/ml) treated in cells was 2.5 mg based on 5 ml of culture medium. Therefore, we used 3 mg KRGE for mouse and 2.5 mg KRGE for astrocyte cells.
In our previous study, astrocytic HIF-1a expression was significantly increased in 0.5 mg/ml KRGE treatment compared to control (0 mg/ml KRGE group) [6] (left figure). In in vivo study, HIF-1a expression in mice brains was much increased in 0.25 mg/ml KRGE administration compared to that in water administration.
- 6. Park, J.; Lee, M.; Kim, M.; Moon, S.; Kim, S.; Kim, S.; Koh, S. H.; Kim, Y. M.; Choi, Y. K., Prophylactic role of Korean Red Ginseng in astrocytic mitochondrial biogenesis through HIF-1alpha. J Ginseng Res 2022, 46, (3), 408-417.
Therefore, the effect of KRGE on HIF-1a expression in vitro and in vivo seems to be similar when we used 0.5 mg/ml KRGE for 24 hr in cell culture and 0.25 mg/ml KRGE for 3 days in mice.

Reviewer 2 Report
The authors show that in astrocyte cultures KRGE increases mitochondrial parameters mass, OXPHOS, O2 consumption. Through a pathway involving: SIRT-3, Hif1alpha and Tom22
Title: The concept "physiologic astrocytes" is hard to accept in an article mainly using cells in culture. Please remove the term "physiologic" from the text. If needed to make a point that it is not under hypoxia, use "normoxia"
The introduction and the scheme in Fig.8c should match more. If something is relevant to be in one, should appear in the other as well. If VEGF is to appear in the scheme, should be measured
Results:
The results are presented in a clear way.
Figure 1. Quantification is needed
Result 3.8 . The text does not match/explain the data presented. Please re-write
Minor comments:
line 102: brain sections could not be 20mm
line 118: the sections must have been observed using fluorescence and not phase contrast
line 225: 30 l of RIPA does not seem right
The grammar and terminology are fine, but the flow and presentation of the information in a coherent manner can be improved.
The introduction should be improved by explaining better the aim of the article and addressing in more detail the relevance to regeneration or aging as it is only mentioned but not linked to the experiments performed.
The description of some of the proteins in the introduction (Nampt, ERR, PDH2) is very detailed, with several unlinked sentences. Some parts could be moved to the discussion to make the research question more clear, as they are later on shown to not play a role in the pathway of interest.
Author Response
Reviewer #2
The authors show that in astrocyte cultures KRGE increases mitochondrial parameters mass, OXPHOS, O2 consumption. Through a pathway involving: SIRT-3, Hif1alpha and Tom22
#1 Title: The concept "physiologic astrocytes" is hard to accept in an article mainly using cells in culture. Please remove the term "physiologic" from the text. If needed to make a point that it is not under hypoxia, use "normoxia"
Response #1: We changed into “Korean red ginseng-induced SIRT3 promotes the Tom22–HIF-1a circuit in normoxic astrocytes”.
#2. The introduction and the scheme in Fig.8c should match more. If something is relevant to be in one, should appear in the other as well. If VEGF is to appear in the scheme, should be measured
Response #2: We measured it. We added VEGF quantification data in Figure 1c. We also changed the Introduction (Line 41-72) and Results (Line 263-265).
Line 41-72:
- Introduction
Astrocytic mitochondria functions appear to be intimately linked to regulation of aging [1, 2], and secretion of angiogenic/neurogenic factors [3-7] under physiological conditions. Since astrocytes boost communication between the vascular and neuronal systems [6, 8, 9], enhanced astrocytes may contribute to dynamic neurovascular communications. Korean red ginseng extract (KRGE) promotes astrocyte proliferation in the subventricular zone, a neurogenic area of adult mice brains [4]. KRG administration increases neurogenesis and the differentiation of neuroblasts in the hippocampus in mice [10]. KRGE can induce angiogenesis both in vivo and in vitro models through activating glucocorticoid receptor [11].
Sirtuins (SIRT 1–7) belong to a family of highly conserved nicotinamide adenine dinucleotide (NAD+)-dependent protein deacetylases. The anti-aging potential of SIRTs has also been studied [12, 13]. Mitochondrial SIRT3 plays a major role in the regulation of a host mitochondrial functions under physiological conditions [14], and the expression thereof has also been detected in neurons and astrocytes obtained from mature rat brains [15].
Increased oxygen consumption from improved mitochondria can stabilize hypoxia-inducible factor-1a (HIF-1a) via proline hydroxylases (PHD1-3) [16-18]. HIF-1a and estrogen-related receptor a (ERRa) act as transcription factor for vascular endothelial growth factor (VEGF) [19, 20], a potent angiogenic and neurogenic factor [21, 22].
Healthy mitochondria have proper mitochondrial membrane potential. Nuclear-encoded proteins are first brought into the mitochondria by the translocase of the outer mitochondrial membrane (TOM) complex in the outer mitochondrial membrane [23]. Tom22 binds to the outer membrane of mitochondria with other TOM complexes, such as Tom40 [24]. Tom20 and Tom22 are essential for maintaining the mitochondrial membrane potential in stressed astrocytes in the presence of KRGE [25].
Our previous study showed that KRGE induced HIF-1a in astrocytes. However, the effects of SIRT3 on HIF-1a-mediated mitochondria functions induced by KRGE have not been well established. We aimed to investigate the relationship between SIRT3 and HIF-1a in KRGE-treated normoxic astrocyte cells, to determine whether SIRT3-mediated mitochondrial functions can stabilize HIF-1a and lead to various biological responses, such as neurovascular regeneration (i.e., neurogenesis, angiogenesis) possibly through VEGF.
Line 263-265:
3.1. KRGE induces VEGF in astrocytes located near the corpus callosum
VEGF is an important growth factor related to regeneration (i.e., angiogenesis, neurogenesis, mitochondria biogenesis) [21, 22, 26, 27].
Results:
The results are presented in a clear way.
#3. Figure 1. Quantification is needed
Response #3: Another reviewer also suggested quantification for fluorescence images. Thank you for your valuable comment. We added quantification graph in Figure 1c, 2b, 2d, 3b, 6b, 6d, 7c, and 7d.
#4. Result 3.8 . The text does not match/explain the data presented. Please re-write
Response #4: We changed it.
Line 425-432:
3.8. KRGE-mediated SIRT3 and Tom22 stabilize HIF-1a protein through PHD2
In the presence of KRGE, si-Tom22-mediated reduction of HIF-1a protein was recovered by co-knockdown of PHD2 (Figure 8a). During normoxia, PHD2 is the key oxygen sensor in the degradation of HIF-1a using oxygen [30, 31]. Similar to Figure 8a, the si-SIRT3-mediated reduction of HIF-1a protein was stabilized by the co-knockdown of PHD2 (Figure 8b). These results suggested that KRGE upregulated mitochondrial SIRT3 and Tom22, leading to mitochondrial activation (i.e., oxygen consumption) and consequent inhibition of HIF-1a degradation.
Minor comments:
#5. line 102: brain sections could not be 20 mm
Response #5: Response: We changed into 20 mm.
#6. line 118: the sections must have been observed using fluorescence and not phase contrast
Response #6: Response: We changed it per your comment.
#7. line 225: 30 l of RIPA does not seem right
Response #7: Response: We changed into 30 ml.
Comments on the Quality of English Language
#8. The grammar and terminology are fine, but the flow and presentation of the information in a coherent manner can be improved. The introduction should be improved by explaining better the aim of the article and addressing in more detail the relevance to regeneration or aging as it is only mentioned but not linked to the experiments performed. The description of some of the proteins in the introduction (Nampt, ERR, PDH2) is very detailed, with several unlinked sentences. Some parts could be moved to the discussion to make the research question more clear, as they are later on shown to not play a role in the pathway of interest.
Response #8: We tried to change Introduction and Discussion per your valuable comments.
Line 439-506: 4. Discussion
KRGE has been known to activate mitochondria functions which can be required for regeneration and anti-aging effects [1, 4, 11, 32-34]; however, it is still unclear what exactly caused the active mitochondrial states it generated. Our previous data showed that an important transcription factor for cell survival and metabolism, HIF-1a, plays a key role in KRGE-mediated astrocytic mitochondrial biogenesis [4]. Our novel findings suggested that mitochondrial astrocytic SIRT3 is an upstream factor for HIF-1a and that the SIRT3–HIF-1a axis regulates KRGE-mediated Tom22 expression. Notably, Tom22 is a crucial factor in mitochondrial oxygen consumption in KRGE-treated astrocytes, leading to stabilization of the HIF-1a protein. Therefore, KRGE boosted the astrocytic mitochondrial function via the SIRT3-mediated Tom22–HIF-1a circuit, leading to enhancement of regenerative factor, VEGF (Figure 8c).
In our observation, HIF-1a may be detected in neurons as well as astrocytes; in addition, we assume that KRGE may increase the HIF-1a levels in neurons. Neuron-specific inactivation of the HIF-1a increases brain injury in a mouse model of transient focal cerebral ischemia [35]. In embryonic neural stem cells, deletion of the HIF-1a gene leads to a decrease in self-renewal of neural stem cells, paralleled by an increase in neuronal differentiation [36]. Therefore, KRGE may have function in neuronal cells by regulating HIF-1a. In this study, the relative prevalence of SIRT3 and VEGF expression in KRGE-administered mice brains can be higher in astrocytes than in neurons localized above the corpus callosum. Thus, we focus on astrocytic VEGF expression through SIRT3–HIF-1a axis.
Nampt-mediated NAD+ biosynthesis is essential for SIRT3 activation [28]. KRGE-induced ERRa expression was downregulated by Nampt knockdown. However, diminished Nampt expression did not downregulate HIF-1a protein levels in KRGE-treated astrocytes. In normoxic conditions, astrocytes treated with KRGE did not result in an increase in the NAD+/NADH ratio. An increase in SIRT3 activation may cause a reduction in NAD+ levels in the mitochondria because the deacetylation activity performed by SIRT3 converts NAD+ into nicotinamide and acetyl ADP-ribose [37]. We hypothesize that KRGE-induced SIRT3 activation may lead to NAD+ reduction, even though KRGE produces NAD+ by Nampt. Our findings suggested that Nampt can regulate ERRa in a SIRT3-independent manner and that SIRT3 can regulate HIF-1a in a Nampt-independent manner.
Energy production is important for brain homeostasis and function. Our previous study demonstrated that KRGE-induced ATP production could be downregulated by si-HIF-1a [4] or si-Tom22 [34]. In this study, KRGE-induced ATP production was completely inhibited by si-SIRT3. KRGE-induced OXPHOS complexes I and IV were reduced with lower SIRT3 expression, indicating that SIRT3 may be implicated in KRGE-mediated mitochondrial OXPHOS and ATP synthesis.
The relationship between HIF-1a and Tom22 in normoxic astrocytes has not been well established. Here, Tom22 may act as an oxygen sensor by stimulating oxygen consumption and HIF-1a stabilization. In contrast, Tom20 did not affect the protein stability of HIF-1a, even though Tom20 knockdown reduced ATP production in KRGE-treated astrocytes [34]. The association between Tom22 and VDAC (known as Por1 in yeast) is important to regulate mitochondrial protein gate assembly [38]. We speculate that the Tom22–VDAC complex-related import system, including Tom40, may be involved in mitochondrial functions, facilitating oxygen consumption. The SIRT3-Tom22 axis regulated the mitochondrial oxygen consumption through PHD2, leading to enhanced HIF-1a stabilization in KRGE-treated astrocytes.
Interestingly, Tom22–HIF-1a circuit influenced the Tom20 expression. As mitochondrial mass can be clearly regulated by the combination of Tom20 and Tom22, the HIF-1a-mediated increase in mitochondrial mass can be modulated by both Tom22 and Tom20. The SIRT3–HIF-1a axis is a critical regulator of Tom20 and Tom22 without affecting ERRa expression under KRGE-treated normoxic conditions.
The peroxisome-proliferator-activating receptor-γ coactivator-1a (PGC-1a)–ERRa pathway is a key regulator for astrocytic mitochondrial biogenesis [3]. The PGC-1a–ERRa pathway also induces HIF-1a stability [17], possibly facilitating VEGF expression [20]. KRGE upregulated ERRa protein levels; however, SIRT3 did not alter ERRa expression. Instead, SIRT1 affected physiological brain functions, including mitochondrial biogenesis in HIF-1a-independent manner, but in an ERRa-dependent manner [34]. Moreover, SIRT4 and SIRT5 regulated both HIF-1a and ERRa. Thus, we cannot rule out the functions of other SIRTs (i.e., SIRT4 and SIRT5) and PGC-1a in KRGE-mediated physiological regenerative effects. Hence, we can investigate the beneficial effects of various SIRTs on KRGE-mediated neurovascular metabolism in the future.
In summary, KRGE enhanced astrocytic VEGF, HIF-1a, and SIRT3 expression near the corpus callosum; however, the exact mechanism by which this occurs is not yet completely understood. According to our novel research, KRGE increased the expression of SIRT3 in astrocytic mitochondria, and SIRT3 promoted mitochondrial biogenesis and ATP synthesis, increasing oxygen consumption. These results triggered the Tom22–HIF-1a circuit through mitochondria-nucleus crosstalk, possibly leading to VEGF upregulation.

Round 2
Reviewer 1 Report
The authors have responded to my major and minor critiques, and I have no further issues.
Author Response
Thank you for your comment.